# Fact :Teaching MLLMs with F̲aithful, C̲oncise and T̲ransferable Rationales

## ABSTRACT

The remarkable performance of Multimodal Large Language Models (MLLMs) has unequivocally demonstrated their proficient understanding capabilities in handling a wide array of visual tasks. Nevertheless, the opaque nature of their black-box reasoning processes persists as an enigma, rendering them uninterpretable and struggling with hallucination. Their ability to execute intricate compositional reasoning tasks is also constrained, culminating in a stagnation of learning progression for these models. In this work, we introduce Fact, a novel paradigm designed to generate multimodal rationales that are faithful, concise, and transferable for teaching MLLMs. This paradigm utilizes verifiable visual programming to generate executable code guaranteeing faithfulness and precision. Subsequently, through a series of operations including pruning, merging, and bridging, the rationale enhances its conciseness. Furthermore, we filter rationales that can be transferred to end-to-end paradigms from programming paradigms to guarantee transferability. Empirical evidence from experiments demonstrates the superiority of our method across models of varying parameter sizes, significantly enhancing their compositional reasoning and generalization ability. Our approach also reduces hallucinations owing to its high correlation between images and text. The anonymous project is available at: https://anonymous.4open.science/r/Fact_program-216D/

## CCS CONCEPTS

• Computing methodologies → Computer vision problems.

## KEYWORDS

Visual Programming, Multimodel Chain-of-Thought, Distillation Step-by-Step

## 1 INTRODUCTION

Multimodal Large Language Models (MLLMs) [5, 7, 19, 21, 44, 45] enhance the natural and sophisticated interaction between humans and machines, offering distinct advantages in solving visual tasks such as image grounding [23], Visual Question Answering (VQA) [16, 29], and scene graph generation [43]. However, these end-to-end large models exhibit an almost black-box level of interpretability: their reasoning processes remain inexplicable, merely leveraging their formidable representational capabilities to fit spurious correlations [30], thus increasing the risk of hallucinations. Furthermore, the compositional reasoning ability of these models is limited [4, 8, 27] (Figure 1). Even the most advanced state-of-the-art (SOTA) proprietary MLLM, such as GPT-4V [32], also struggles to perform well in tasks involving counting [1], spatial reasoning [24] and intricate VQA tasks [16, 29].

In this paper, our objective is to enhance the explicit intermediate reasoning capabilities of MLLMs by distilling interpretable rationales. An ideal rationale should embody three properties: 1)

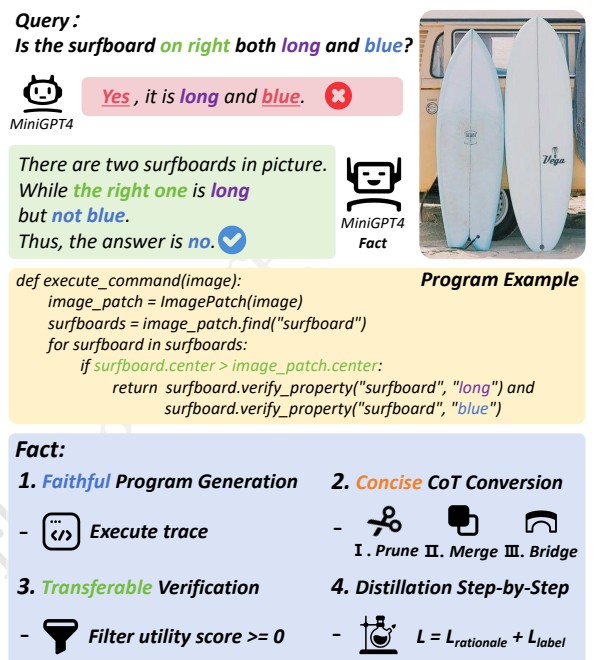

**Figure 1: MLLMs exhibit limited proficiency in combinatorial reasoning and spatial understanding. While Fact can significantly enhance their capabilities in performing visual tasks.**

**Faithfulness**. Faithfulness necessitates that the rationale's reasoning process remains staunchly aligned with the model's conclusions. This alignment is paramount as it ensures the integrity and reliability of the reasoning process, ultimately leading to conclusions that are both logical and accurate. However, verifying the faithfulness of a rationale poses significant challenges. It is not uncommon for rationales to resemble logical reasoning superficially [33], yet upon closer inspection, they may lack a direct correlation to the intended conclusion [40], undermining the model's effectiveness and trustworthiness. 2) **Conciseness**. The principle of conciseness stresses the importance of eliminating superfluous in-context information that does not contribute to the reasoning process. The presence of irrelevant information can obfuscate the model's reasoning pathway, leading to reduced decision-making accuracy [17, 36]. Extracting a concise and clear rationale is still a process fraught with difficulty, yet such brevity aids in efficiency. 3) **Transferability**. Transferability refers to the ability of a rationale to encapsulate the model's explicit inductive understanding of knowledge in a manner that is applicable across various models, paradigms, and scales. A transferable rationale enhances the generalizability of the model's learned reasoning [28], facilitating the sharing of insights and improvements across different models. However, achieving a high degree of transferability is challenging due to the diverse architectures and learning mechanisms employed by different models.

However, obtaining rationales that satisfy the aforementioned properties is a challenging endeavor. Firstly, the high costs and low efficiency of manually annotating rationales make this approach impractical for large-scale applications. Furthermore, relying on templated rationales [20, 30, 46], while beneficial for specific tasks, severely limits adaptability and scalability. Their rigid structure precludes the possibility of nuanced adaptation, confining their utility to narrowly defined scenarios. Moreover, the use of scene graphs [13] and neural symbols[33, 42], although innovative, demands a level of understanding and utilizing external tools that not all models possess. This approach is hindered by its poor transferability. Given these limitations, there is an urgent need to improve the quality of rationale to meet its diverse requirements effectively.

In this work, we explore a different novel method: **Fact**, as illustrated in Figure 1, leveraging the powerful symbolic reasoning capabilities of code-pretrained models to construct a faithful, concise, and transferable natural language rationale, which is then applied in teaching MLLMs. Specifically, 1) we **generate faithful code** by employing a code generation model [31] to utilize its compositional capabilities of various tools for the visual task. Then we record the execute trace as a draft chain-of-thought (CoT), retaining only those snippets that compile successfully and yield correct outcomes. 2) We define three operations on the rationales: **pruning, merging, and bridging**, to simplify code execute trace into natural language by pruning irrelevance in abstract syntax tree (AST), merging duplicates in symbolic traces, and bridging logical gaps to form coherent CoT. A language model [31] is then utilized to seamlessly bridge these gaps and refine the narrative. 3) We **filter transferable rationales** that successfully transfer programmatic reasonings to end-to-end models [3, 47] and 4) **distillate the rationale** step-by-step [14, 34], aiming to enhance the quality and applicability of distilled knowledge across various domains.

In our paradigm, we first ensure that the rationales are **correct and rigorous**: Given that programs can accurately guide the derivation of answers, the rationales based on these programs can also faithfully represent both the reasoning process and its outcomes [37]. Secondly, our rationales are **concise and brief**: the unexecuted branches within programs are pruned, the repetitive assignments in loops are merged, the repeated uses of tool combinations are inductively summarized, and the gaps in the reasoning process are bridged. Thirdly, our rationales are **transferable and consistent**: the CoT rationales derived from generated from programming execution traces are selectively retained, only emphasizing those that aid end-to-end models in achieving the correct answers through logical reasoning.

Our experiments demonstrate that CoT rationales, characterized by faithfulness, conciseness, and transferability, generally enhance MLLM performance on downstream tasks [1, 16, 23, 29]. Trained with these rationales, MLLMs not only show robust capabilities in counting and compositional reasoning tasks but also guide large vision models to comprehend and reason more logically. The directness of the rationales, devoid of extraneous irrelevance, effectively reduces the occurrence of hallucinations. Ablation studies within our research framework further substantiate the indispensability of these three attributes, affirming the superiority of our method from multiple perspectives.

Accordingly, the contributions are delineated as follows:

- We introduce an innovative paradigm, **Fact**, to distill code-pretrained models' logic into a faithful, concise, and transferable rationale for teaching MLLMs, enabling them to harness advanced reasoning capabilities.
- We refine CoT rationales through controllable editing operations—pruning, merging, and bridging—to enhance their quality and coherence. Furthermore, we have also trained a compact model to identify logical gaps. These operations effectively address challenges in maintaining relevance in distilled knowledge.
- We validate that the CoT rationale generated through a programming-based approach is applicable for distillation into end-to-end models. It underscores the practicality of our CoT rationales generation paradigm, showcasing its versatility and effectiveness in enhancing model learning.
- Highlighting the versatility and transferability of our approach, we demonstrate its capability to enhance the reasoning abilities of MLLMs across various sizes. Our evaluation on tasks like GQA [16], OKVQA [29], TallyQA [1], and COCO [23] reveals significant performance improvements in MLLMs, and effectively reduces hallucinations by being devoid of extraneous irrelevance.

## 2 RELATED WORK

### 2.1 Visual Program Distillation

Visual programming is a burgeoning field that employs neural symbols [12] or Python modules [38] for task synthesis and execution. While it offers enhanced performance and interpretability through precise image manipulation via code, its dependency on multiple models and prolonged inference times necessitates substantial computational resources. In contrast, our approach and Visual Program Distillation (VPD) [15] diverge significantly in handling program-based methods. VPD simplifies multimodal learning by distilling tool use and programmatic reasoning into smaller models but retains unnecessary execution traces, lacks precise spatial task execution, and overlooks the verification of transferability. Our method addresses these limitations through Efficient Rationale Editing: Unlike VPD, we implement controllable editing to refine programs into concise rationales; Enhanced Spatial Task Execution: Our approach exhibits superior spatial reasoning capabilities, enabling more accurate completion of spatial tasks compared to VPD; Verified Transferability: We also emphasize and validate the transferability of program-based rationales to end-to-end models which VPD simply ignored it. These distinctions underscore our contribution to integrating complex reasoning within MLLMs.

### 2.2 Rationale Distillation

Deploying LLMs in practical applications is challenging due to their substantial memory and computational demands. A feasible approach to mitigate these challenges is training smaller, task-specific models via fine-tuning or distillation with labels generated by LLMs. While effective, these strategies often necessitate extensive training data to match the performance with LLMs [25, 26]. As a solution, Distillation Step-by-Step [14] emerges, enabling smaller models to potentially surpass LLMs with less training data. However, existing efforts rarely account for the quality of distillation data and

### 1. Faithful Program Generation

**Query:**
*How many people are wearing the hat?*
**Image:**

🤖 *Code Pre-trained Model:*

```
def execute_command(image):
    image_patch = ImagePatch(image)
    person_patches = image_patch.find("person")[1]
    hat_patch = image_patch.find("hat")[0]
    person_in_hat = 0
    for person_patch in person_patches:
        if hat_patch.overlaps_with(person_patch):[2]
            person_in_hat += 1
    return person_in_hat
```

[1] The word "people" will be replaced by "person" in the **find** function for higher accuracy.
[2] The expression should be **overlaps_with**(patch.left, patch.lower, patch.right, patch.upper).

### 2. Concise CoT Conversion

**Prune:**

**False** ✂️
.overlaps_with( )

**True**
.overlaps_with( )

**Merge:**
{person_in_hat : 0}
{person_in_hat : 1}   ⬛ *reassign*

**Bridge:**
*Two persons (22,57,80,150) (83,58,140,148) in the picture. <no-gap> ✔*
*One hat (96,112,119,148) in the picture. <no-gap> ✔*
*The person (83,58,140,148) is overlap with the hat (96,112,119,148). <no-gap> ✔*
*The number of person in hat is 1.*

### 3. Transferable Rationale Verification

**Rationale:**
*There are two people at (22,57,80,150) (83,58,140,148) and one hat at (96,112,119,148). Upon detection, the person at (83,58,140,148) is wearing the hat (96,112,119,148). Thus, the number of people wearing a hat is 1.*

**Input:**

| Query Image | Query Image + Rationale |
| --- | --- |

🤖🤖 *End-To-End Models*

**Output:**

| 2 | 1 | ✔ *Useful* |
| 1 | 1 | ✔ *Unsure* |
| 2 | 2 | ✘ *Non-Useful* |

**Gold Answer = 1**

### 4. Distillation Step-by-Step

**Rationale fine-tuning:**

| Query Image + Rationale | Query Image + Label |
| --- | --- |

🤖 **MLLM**

*There are ...*
*Upon ...*                     **1**
*Thus, ...*

$$L = L_{rationale} + L_{label}$$

**Output:**

🤖 **MLLM-Fact**

**Figure 2: The pipeline of Fact: 1) Generate executable code from an image and query using a code generation engine and retain code that correctly reasons against expected answers. 2) Simplify code into natural language by pruning irrelevant AST nodes, merging duplicates in symbolic traces, and filling logical gaps to form coherent CoT. 3) Evaluate and filter CoTs for end-to-end model feasibility. 4) Distill refined, accurate CoTs into MLLMs for enhanced adaptability.**

its transferability between large and small models. This neglect can result in the distillation process incorporating irrelevant information, which diverts the smaller model and detracts from its performance. Our approach emphasizes the quality and relevance of distilled information, tackling the common pitfalls of distillation. Furthermore, our method stands out by offering verifiability and transferability. This ensures that the distilled knowledge is not only accurate and relevant but also adaptable across different models and tasks. The ability to verify the correctness and applicability of our CoT rationales sets our work apart, underscoring its novelty and effectiveness in the context of distillation methodologies.

## 3 METHOD

To furnish MLLMs with faithful, concise, and transferable rationales, we introduce **Fact**, a comprehensive, model-agnostic paradigm for generating, editing, and filtering precise CoT rationale framework, as depicted in Figure 2. Given an image and a corresponding query, we first generate executable code and derive an output, comparing it against the expected answer and retaining the code that leads to correct reasoning (Section 3.1). To transfer the programming language and its embedded knowledge to MLLMs in the form of natural language, we undertake a series of operations: pruning irrelevant nodes from the AST, merging the duplicate items in symbolic traces, and bridging language expressions with logical gaps, thereby generating linguistically and logical coherent CoT rationales (Section 3.2). To ensure that the CoTs generated via the programming-based paradigm are feasible for training end-to-end models, we evaluate them by end-to-end models for effective filtration and verification (Section 3.3). Ultimately, we can get accurate, non-redundant CoT rationales that are adaptable to various end-to-end models, and ready to be distilled into MLLMs (Section 3.4).

### 3.1 Faithful Program Generation

The reason for utilizing the execution trace of visual programming as the initial CoT rationale lies in the fact that the execution trace faithfully leads to its own answer. A generated program itself embodies a distilled manifestation of a large language model's capacity to amalgamate and apply knowledge. Furthermore, the adherence of code output to established syntactical rules allows for the verifiability of the program's authenticity. Thus, programming-based rationale that accurately produces the correct answer guarantees the fidelity of its reasoning process.

More specifically, for a given image $x$ and a corresponding query $q$, we employ a program generator $\pi$ (GPT-3.5-turbo [31]) to generate code $z = \pi(x, q)$, the necessary APIs and tool models involved during this process we followed the same configuration with the practices established by ViperGPT [38]. However, distinct from these methodologies, we have re-engineered a new interpreter $\phi$, which dynamically edits the code during execution to manage the trace $t = \phi(z, x)$. We expound on these operations in Section 3.2. Due to Python's extensive array of built-in functions and logical statements, the execution of a program mirrors the process of solving a problem, thereby ensuring the relevance and accuracy of the context, leading to the correct outcome. By comparing the program's output against expected answers, we identify and preserve the correct instances as candidate solutions for further analysis.

Our approach emphasizes fidelity by integrating the interpreter's logical capabilities with the perceptual strengths of pre-trained models. This synergy ensures that the generated CoTs are not only interpretable but faithful to the represented logic, mirroring the reliability of their conclusions. Thus, **Fact** can guarantee that CoTs reflect a true and faithful rationale, maintaining the integrity and authenticity of the underlying reasoning process.

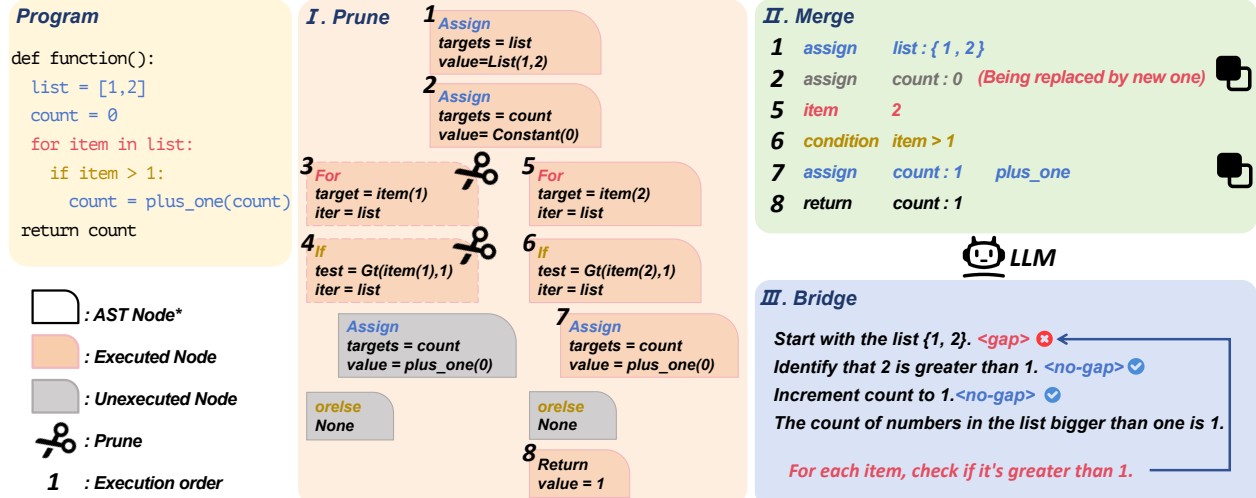

**Figure 3: We use a Python program to explain our editing operation: I) Parse executed code lines into corresponding AST nodes and prune unused loops and conditions, organizing output into a symbolic trace. II) Merge iterated outputs and update variables, converting the symbolic trace to natural language using an LLM. III) Train a small model to identify gaps between statements, filling it with an LLM to complete the logic of the CoT rationale for clarity and coherence.**

## 3.2 Concise CoT Conversion

Directly utilizing code for distillation learning is not advantageous due to redundancies within the code, like unmet conditionals in *if* statements, and repetitive outputs from *for* loops. Previous work [17, 36] demonstrated that models are prone to distraction by irrelevance. Moreover, for models not trained on code corpora, understanding programming becomes significantly challenging. In light of this, we propose three operations to transform program traces into concise CoT rationales: dynamic pruning, symbolic merging, and logical bridging. These operations aim to refine the generated CoTs by removing irrelevance, combining repetition for conciseness, and bridging gaps in logicen for coherence, thereby enhancing the quality and relevance of the distilled content for MLLMs.

**Dynamic Pruning**: The reason for taking the pruning operation is that the rationale should explain why it is, rather than why it is not. To this end, we initially construct the code $z$ into an AST, denoted $t = \{V, E\}$, where $V = \{v_1, ..., v_n\}$ represents the vertices and $E = \{e_1, ..., e_m\}$ represents the edges. The AST facilitates more granular modifications via node editing compared to direct manipulations of the code text alone. Given a specific image, corresponding query, and the generated program, the final output answer will be unique. Thus, variables and code statements that are not used in the execution process should be discarded. For example, within conditional statements (*if*), if the condition is not met, we should not record this condition as false, since the program does not enter the scope of the if statement. By doing this, we can remove content that is irrelevant to the context of the reasoning process.

**Symbolic Merging**: The rationale we envisage should exhibit inductive capabilities rather than merely iterative repetitions. Unfortunately, programs are inherently adept at executing loops but typically lack the capacity for induction. To mitigate this, we edit each node preserved through the pruning process, encoding the output as a symbolic trace based on a defined schema:

- **Operation**: The attributes of different operation nodes in AST represent their corresponding operations, such as assign, function, loop, and condition. We can modify them through the "*generic_visit*" functions or "*visit_If*" etc.
- **Arguments**: We use a dictionary to store the variables and their names involved in the operation process.
- **Invocation**: If there is a function or Python tool utilized within an operation, it is noted under invocation, which is an optional component.

For instance, a code line like "*num = len(patches)*" would be recorded after execution as "*assigned num:8 len*". This approach allows for the effective tracking and overlaying of repeated variables after program execution, retaining only the last symbolic trace of a variable. Such a method significantly curtails redundancy by eschewing the retention of all outputs, thereby streamlining the CoT trace into a more coherent and concise CoT rationale.

**Logical Bridging**: After the processes of pruning and merging, we cannot guarantee that all the retained logical relationships remain intact. Consequently, directly using text CoT transfer from symbolic trace by language models as distilled data is impractical. To address this, we have specially trained a lightweight network to determine whether a gap exists between sentences. This was achieved by fine-tuning a T5 [6] model, leveraging *articles* from the DROP dataset [9], renowned for its logical reasoning relational content, as positive instances. In contrast, *articles* with their sentence order arbitrarily shuffled served as negative instances. If it is the relationship between the preceding and following sentences, a <no-gap> tag is inserted between them; otherwise, a <gap> tag is used to indicate the logical discontinuity. It is important to note that we do not employ any other manually annotated labels or additional tags from the DROP [9] dataset, thus avoiding the risk of data leakage.

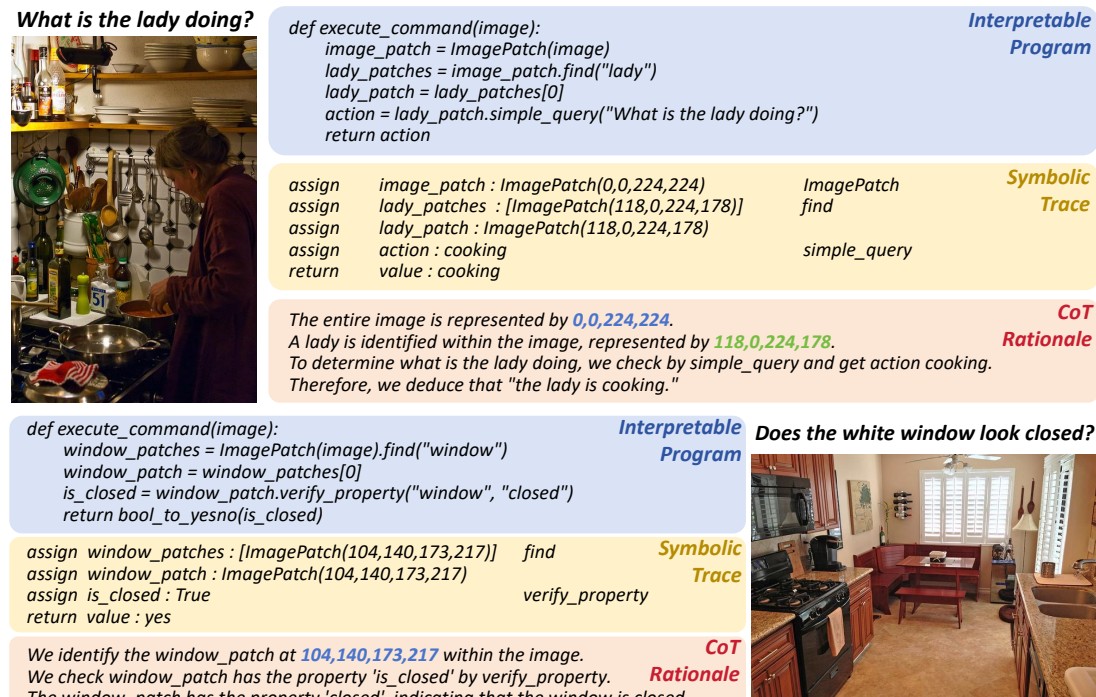

**Figure 4: We show several examples of the process that generates CoT rationale for distillation.**

Ultimately, we employ a language model to not only fill in the gaps within sentences but also to significantly enhance the underlying reasoning process. This refined the original content, culminating in the development of a concise and coherent CoT.

## 3.3 CoT Transferability Verification

Previous research [18] has investigated whether rationales generated by machines remain applicable to humans, yet there has been no exploration into whether the reasoning paradigms of visual programming succeed in transferring knowledge to end-to-end vision models. We define the improvement that rationales brought to untrained models as their **utility score**, which serves as a measure for evaluating the transferability of CoT rationales, especially for those rationales generated by visual programming transferring to end-to-end models. Historically, the process of directly distilling CoTs into end-to-end downstream models proceeded without considering whether it was appropriate for the student model. Our research aims to address this oversight by ensuring that distilled rationales not only remain pertinent but also significantly boost the capabilities of MLLMs.

We adopt the same categorization as GEN-U [18], namely Useful, Non-Useful, and Unsure. Specifically, for end-to-end models like OpenFlamingo [3] MiniGPT4 [47] and mPlug-Owl [44] if they previously failed to solve a task correctly and after the introduction of a rationale corrects their answers, then the rationale is deemed *Useful* (**+1 score**). Conversely, if the models continue to solve the task incorrectly even after the rationales have been presented, this indicates that the rationale is *Non-Useful* (**-1 score**). Furthermore, if the models correctly solve the task both before and after the rationale is demonstrated, we cannot definitively determine the role of the

rationale in aiding task resolution. We refer to these rationales as *Unsure* (**+0 score**). Ultimately, we will provide programming-based rationales for these end-to-end models and retain the rationales whose score is greater than or equal to 0 as the final knowledge taught to the MLLMs.

This approach ensures a focus on rationales that potentially enhance the performance of student models in machine learning by evaluating their usefulness thereby improving knowledge transferability between models.

## 3.4 Distillation Step-by-Step

In our approach, instead of using rationales as additional model inputs, we frame learning with rationales as a multi-task problem, we adopt the same loss function as Distillation Step-by-Step [14] to train the model with both label and rationale as input. In other words, the $f(x, q) \rightarrow \hat{y}$ and $f(x, q) \rightarrow \hat{r}$ are trained with:

$$L_{label} = \frac{1}{N} \sum_{i=1}^{N} l(f(x_i, q_i), \hat{y}_i)$$

$$L_{rationale} = \frac{1}{N} \sum_{i=1}^{N} l(f(x_i, q_i), \hat{r}_i)$$

, where $\hat{r}$ represents the Cot rationales corresponding to picture $x$ and question $q$, and $\hat{y}$ represents the labeled answer of the data set. This formula enables it not only to predict the task labels but also to generate the corresponding rationales given the text inputs. Thus, the loss function is formulated as :

$$L = L_{label} + \lambda L_{rationale}$$

Table 1: Compare **Fact** with per-train MLLMs and corresponding *instruct* model on zero-shot benchmarks.

| | Language Model | COCO | Flickr 30K | VQAv2 | GQA | OK-VQA | TallyQA Simple | TallyQA Complex |
|---|---|---|---|---|---|---|---|---|
| CosMo (2B) [41] | OPT-IML-1.8B | 79.9 | 51.3 | 46.7 | - | 28.3 | - | - |
| Flamingo (3B) [2] | Chinchilla-1.4B | 73.0 | 60.6 | 49.2 | - | 41.2 | - | - |
| OpenFlamingo (3B) [3] | MPT-1B | 74.9 | 52.3 | 44.6 | 30.1 | 28.2 | 64.4 | 59.3 |
| OpenFlamingo-*Instruct* (3B) *generalist* | MPT-1B | 79.7 | 53.8 | 45.9 | 30.9 | 30.3 | 65.9 | 61.8 |
| OpenFlamingo-**Fact** (3B) *generalist* | MPT-1B | **85.3** | **56.6** | **49.2** | **32.4** | **31.8** | **70.1** | **65.7** |
| OpenFlamingo-*Instruct* (3B) *specialist* | MPT-1B | - | - | 47.7 | 32.6 | 31.7 | 70.1 | 66.5 |
| OpenFlamingo-**Fact** (3B) *specialist* | MPT-1B | - | - | **51.0** | **35.5** | **35.7** | **77.6** | **68.0** |
| VL-GPT (7B) [48] | LLaMA-7B | 116.4 | - | 51.7 | 34.6 | 35.8 | - | - |
| BLIP-2 (7B) [21] | Vicuna-7B | - | 74.9 | 65.0 | 41.0 | 45.9 | - | - |
| MiniGPT4 (7B) [47] | Vicuna-7B | 99.6 | 76.3 | 46.9 | 34.5 | 35.1 | 69.5 | 60.5 |
| MiniGPT4-*Instruct* (7B) *generalist* | Vicuna-7B | 105.5 | 78.5 | 48.2 | 34.9 | 36.9 | 71.3 | 63.8 |
| MiniGPT4-**Fact** (7B) *generalist* | Vicuna-7B | **116.8** | **83.7** | **50.8** | **36.6** | **38.3** | **74.4** | **66.9** |
| MiniGPT4-*Instruct* (7B) *specialist* | Vicuna-7B | - | - | 51.1 | 37.2 | 40.6 | 75.2 | 67.2 |
| MiniGPT4-**Fact** (7B) *specialist* | Vicuna-7B | - | - | **54.2** | **39.8** | **42.0** | **80.7** | **71.3** |

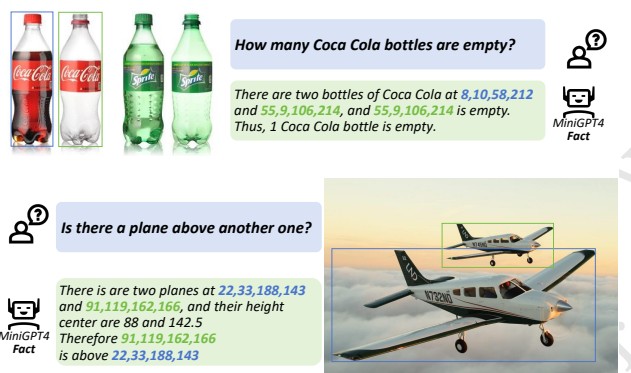

**Figure 5: Example outputs of MiniGPT4 trained with Fact.**

We set $\lambda$ to 1 to ensure that both the prediction of labels and the generation of rationales are equally prioritized. This balanced approach underscores our commitment to fostering a model that is proficient in both accurate prediction and the articulation of coherent, logical rationales.

In the training of multimodal models, ensuring a precise alignment between textual outputs and specific image regions has presented challenges across existing methodologies. The crux of the issue lies in two main factors: 1) Images are commonly resized to a uniform dimension of 224x224 pixels during training, and 2) No predefined regions or IDs are available to assist in the object localization process before training. Addressing these challenges, our approach simplifies the task by resizing all images to 224x224 pixels prior to processing. This standardization reduces the complexity associated with variable image sizes, allowing the model to focus solely on enhancing its alignment capabilities without the need for additional external information. By adopting this method, we effectively overcome pixel displacement issues that occur with scaling and normalization, thus achieving a more accurate correspondence between text descriptions and image regions.

## 4 EXPERIMENT

**Fact** is a model-agnostic paradigm capable of generating faithful, concise, and transferable rationales, thereby teaching MLLMs effectively. In this section, we outline our experimental setup (Section 4.1). Then, we conduct experiments on various zero-shot comprehension tasks. (Section 4.2). We are also curious about whether the rationale output by the specialist model can help the general large model excavate the details, and explore the mutual reinforcement performance of this process between large and small models (Section 4.3). Extensive ablation studies were also undertaken to further examine the contributions of different components of our approach (Section 4.4). This comprehensive experimental framework is designed to thoroughly assess the efficacy and versatility of the **Fact** paradigm in improving the performance of MLLMs.

### 4.1 Experimental Setup

**Model Setup**. We compare **Fact** against two models with different parameter sizes as backbones: MiniGPT4 with Vicuna 7B [47] and OpenFlamingo 3B [3]. In generating rationale, we utilize GPT-3.5-turbo [31] as our code generation model. In visual programming, we adopt the same configuration and APIs as used in ViperGPT [38]. LLama2-70B [39] serves as the bridge language model, and we fine-tune a T5 [6] model to identify logical gaps.

**Training Settings**. We trained two versions of the backbone and a control model respectively:

- The *generalist* model employs multi-task training with rationale distillation to demonstrate the general utility of **Fact**.
- The *specialist* model further training on specific task to demonstrate **Fact**'s adaptation to the task.
- We also employ a control model using the same data and parameter for fair comparison called *Instruct*.

For the *generalist* model, we employ multi-task training to fine-tune a pre-trained model. This fine-tuning process encompasses a broad array of subsets from both image captioning, including COCO [23] and Flickr [35] captions, and VQA tasks. The latter includes

**Table 2: Comparison of MME and POPE benchmark.**

|  | MME | POPE | | |
|---|---|---|---|---|
|  |  | Random | Popular | Adversarial |
| OpenFlamingo [3] | 668.2 | 52.6 | 67.2 | 56.0 |
| w/ *Instruct generalist* | 847.3 | 69.5 | 73.1 | 68.4 |
| w/ **Fact** *generalist* | 912.2 | 73.0 | 75.6 | 71.5 |
| MiniGPT4 [47] | 581.7 | 43.3 | 50.8 | 47.9 |
| w/ *Instruct generalist* | 864.9 | 68.3 | 74.4 | 71.2 |
| w/ **Fact** *generalist* | 1034.7 | 78.7 | 83.7 | 79.1 |

general VQA (VQAv2 [11]), compositional questions and reasoning (GQA [16]), counting (TallyQA [1]), and VQA that requires external knowledge (OK-VQA [29]). These tasks present a textual input alongside an expected label output. Utilizing the **Fact** pipeline, we synthesize CoT rationales for these labels, subsequently fine-tuning the backbone model based on the loss described in Section 3.4. For half of VQAv2 data and image captioning tasks that do not necessitate the generation of a program for captioning, we set the $L_{rationale}$ to 0, maintaining only $L_{label}$.

The *specialist* model undergoes a training process identical to that of the generalist model, with the key distinction being the replacement of the general training dataset with the specific training set corresponding to the task at hand. This substitution aims to enhance the model's performance on particular tasks. The training code for both models is provided in the supplementary material for further reference.

For the *instruct* models, we exclude the CoT rationale from our data mixture using identical parameters and procedures for both generalist and specialist models. This experimental setup was designed to clarify any misconceptions regarding the efficacy of instruct tuning alone, thereby demonstrating that the observed improvements in model performance are specifically attributable to the enhanced composite reasoning and spatial understanding capabilities provided by high-quality CoT rationales.

In the process of generating CoT rationales, we implemented filtering twice, focusing on faithful selection within the programming portion and transferability selection for the CoT rationales. The total number of samples is detailed in Appendix A for reference.

**Baselines**. The paper also lists results from other multimodal models like CosMo 2B [41], Flamingo 3B [2], VL-GPT 7B [48] and BLIP-2 7B [21] for comparison.

**Benchmark**. Our models are rigorously evaluated on a comprehensive suite of zero-shot benchmarks to ascertain their performance across various tasks. Specifically, for image captioning, we utilized datasets from COCO [23] and Flickr 30K [35], with the model's performance assessed using the CIDEr scoring metric. For VQA, we employed the VQAv2 [11] dataset, evaluating the model on the test-dev split. The GQA [16] benchmark was assessed on its test-dev set, while the OK-VQA [29] was evaluated on the val split. For counting tasks, we used the TallyQA [1] benchmark, assessing our model on the test set. In addition to these primary benchmarks, we extended our evaluation to include additional benchmarks such as the MME [10] score and POPE [22] F1-score, which are designed to further test the model's capabilities in multi-modal understanding and predictive object positioning, respectively. We present the prompts required for evaluating downstream tasks in Appendix B.

**Table 3: Use the model's output as a rationale prompt for a large vision model and test at GQA task.**

| mPlug-Owl (13B) *pre-trained* | GQA | OK-VQA |
|---|---|---|
| + in-context prompt | 56.5 | 57.6 |
| + CoT (OpenFlamingo-**Fact**) | 62.7 | 62.6 |
| + CoT (MiniGPT4-**Fact**) | 63.2 | 63.0 |

## 4.2 Quantitative Results

In this section, we evaluate the performance of two distinct models characterized by varying parameter magnitudes: MiniGPT4 [47], equipped with Vicuna 7B, and OpenFlamingo 3B [3], after undergoing rationale training and compare them with other pre-train models in Table 1. Examination of the *generalist* model reveals that post CoT rationale distillation, there is an observable enhancement in general performance, substantiating the hypothesis that MLLMs can indeed derive substantial benefits from such distillation processes. For *specialist* models, in tasks requiring compositional reasoning, such as GQA [16], and counting tasks, such as TallyQA [1], **Fact** outperformed *instruct* by 2.6%, 5.5%, and 4.1%, respectively. These results indicate a significant enhancement in the model's understanding of counting and mastery of logic. Such capabilities are largely attributed to the spatial understanding and tool integration abilities provided by high-quality rationales.

Our additional benchmarks, as presented in Table 2, demonstrate that **Fact** enhances the perception and cognition capabilities of MLLMs and reduces hallucinations by refining rationales to include only objects relevant to the question. This rationale significantly improves the relevance between text and images, showcasing **Fact**'s capacity to direct MLLMs' focus towards pertinent details and thereby increase accuracy. This enhanced focus not only optimizes model performance but also underscores the critical role of tailored rationale design in achieving precise model responses.

## 4.3 Migrating to Large Models

We posit that CoT rationales generated by the student model could mutually benefit the teacher model as well. To test this theory, we devised an experimental framework that applied two specific types of contextual prompts to the mPlug-Owl 13B [44] model in the GQA and OK-VQA context: 1) a direct question followed by its answer, and 2) a question accompanied by CoTs produced by task-specific models, plus the answer, with a consistent presentation of three prompts. The outcomes observed on both tasks on test sets are compiled in Table 2. These findings corroborate our assertion that CoTs do indeed bolster the inferential precision of teacher models. Interestingly, variations in performance enhancements attributed to CoTs sourced from MiniGPT4 and OpenFlamingo were minimal when applied to the larger model. This phenomenon likely stems from the extensive comprehension capabilities inherent to larger models, enabling them to effectively leverage the supplied CoTs for detailed and accurate deductions. This not only validates the utility of CoTs across model scales but also highlights the adaptability of larger models to assimilate and refine input from smaller counterparts, further emphasizing the symbiotic potential between models of differing capacities.

**Table 4: Ablation results (%) of individual components.**

|   |                  | GQA  | OK-VQA |
|---|------------------|------|--------|
| 0 | Backbone         | 30.1 | 28.2   |
| 1 | + faithfulness   | 30.8 | 29.7   |
| 2 | + conciseness    | 31.9 | 31.5   |
| 3 | + transferability| 32.4 | 31.8   |

## 4.4 Ablation Experiment

**Qualitative Analysis**. We begin by presenting several qualitative examples in Figure 4 to illustrate our process from visual programming to symbolic trace, and subsequently to multimodal CoT rationales. Throughout progression, we continually refine the capabilities of large models into more comprehensible language, ultimately distilling this knowledge into MLLMs. This approach demonstrates the efficacy of our method in harnessing the sophisticated reasoning abilities of teacher models and making them accessible to MLLMs through a distilled, understandable format.

**Analysis on Fact**. For the three properties of CoT proposed in our paper, we conducted sequential ablation experiments using OpenFlamingo (3B) as the backbone, with results on the GQA documented in Table 4: 1) Backbone: Merely use a language model to convert the entire execution trace into CoTs for distillation. 2) Backbone + faithfulness: Here, we filter the program and retain pieces of code that produce the correct answers, thereby ensuring faithfulness. 3) Backbone + faithfulness + conciseness: At this stage, operations such as pruning, merging, and bridging are performed during execution to enhance conciseness, eliminating redundancy. 4) Backbone + faithfulness + conciseness + transferability: Finally, we focus on transferability, selecting rationales that are most suitable for distillation. The results demonstrate that each of the three properties significantly contributes to performance enhancement. Faithful CoTs enable the model to infer correct outcomes, conciseness helps the model focus more on reasoning logic, and transferability effectively filters data suited for distillation, achieving a more suitable knowledge representation for smaller models.

**Analysis on CoT edition**. We also evaluated the three CoT rationale editing operations on OpenFlamingo 3B. This particular part of our experiment retained the comprehensive method of faithfulness and transferability, with the only difference in the CoT editing aspect. The outcomes of this detailed analysis are presented in Table 5. Among the editing strategies, pruning exhibited the most profound effect on enhancing model performance, with merging and bridging following in order of impact. Our analysis suggests that pruning effectively removes extraneous information that could detract from the model's focus, thus facilitating the most substantial gains in performance. Merging, aimed at simplifying CoTs by integrating repeated variables based on predefined criteria, yielded marginal improvements. Its impact is less prominent attributed to the nature of certain tasks within our experiment, which did not necessitate the amalgamation of variables. Bridging, engaged towards the end of the editing process, fine-tunes the CoTs for better linguistic flow and logical sequencing, albeit making only slight contributions to the overall efficacy of the CoTs. This sequential refinement underscores the importance of targeted editing in optimizing CoTs for both clarity and efficiency in reasoning.

**Table 5: Ablation results (%) of CoT rationale edition.**

|   |                     | CoT rationale edition | | | Accuracy | |
|---|---------------------|-------|-------|--------|------|--------|
|   |                     | prune | merge | bridge | GQA  | OK-VQA |
| 0 | Backbone            |       |       |        | 30.9 | 29.9   |
| 1 | + prune             | ✓     |       |        | 31.7 | 31.4   |
| 2 | + merge             |       | ✓     |        | 31.1 | 30.6   |
| 3 | + bridge            |       |       | ✓      | 31.1 | 30.4   |
| 4 | + prune + merge     | ✓     | ✓     |        | 32.2 | 31.7   |
| 5 | + prune + bridge    | ✓     |       | ✓      | 32.0 | 31.5   |
| 6 | + merge + bridge    |       | ✓     | ✓      | 31.4 | 31.0   |
| 7 | OpenFlamingo-**Fact** | ✓   | ✓     | ✓      | 32.4 | 31.8   |

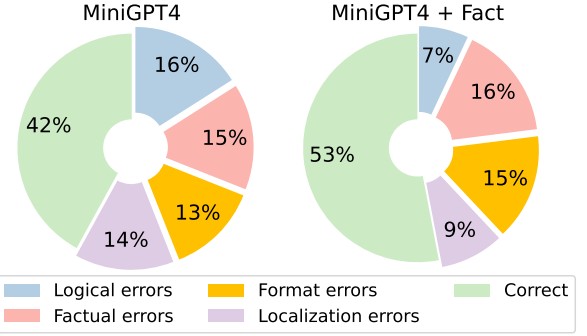

**Figure 6: Sources of error in GQA task.**

**Human Evaluation** We manually chose 100 responses from the GQA dataset and manually undertook a detailed error analysis. The errors identified were classified into four main categories: Logical, Factual, Format, and Localization errors. We will detail the types of errors in Appendix C. For MiniGPT4, we utilized the prompt, "Answer the following VQA questions step by step." Subsequently, we juxtaposed these outcomes with those derived from using **Fact**, illustrating the comparative analysis in a graphically represented figure 6. A noteworthy observation is a pronounced decrease in both Logical errors and Localization errors. This is consistent with our expectation that **Fact** can improve the reasoning and spatial capabilities of MLLMs. However, we conjecture that constraints inherent to the model parameters contributed to the persistence of Format errors across both methodologies. This suggests that while our approach significantly mitigates certain types of errors, the challenge of completely eliminating format-related inaccuracies remains, indicating a potential area for further refinement in model training and prompt engineering strategies.

## 5 CONCLUSION

This study presents **Fact** for generating, refining, and distilling CoT rationales to enhance multimodal models' reasoning capabilities. Through targeted CoT editing operations such as pruning, merging, and bridging, we efficiently remove irrelevant information and improve the coherence of CoTs. By teaching these high-quality rationales to MLLMs, we significantly boost model performance across various tasks. Our experimental results underscore the importance of precise CoT rationale refinement, demonstrating marked performance enhancements in both teacher and student models.

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
