# OpenReview forum: "Fact: Teaching MLLMs with Faithful, Concise and Transferable Rationales"
_acmmm.org/ACMMM/2024/Conference — MM2024 Poster_

### Official Review · Reviewer_7FGt · 2024-05-02

**Rating:** 5
**Confidence:** 2

**Summary:**

This paper proposes a framework to generate, refine and distill rationales from code pre-trained models and teach MMLMs to understand these rationales while providing final answers. The experiments demonstrate the effectiveness of Fact on various VL understanding tasks when integrated into existing MMLMs.

**Strengths:**

1. The presentation is easy-to-read and clear. The logic is smooth.

2. The idea of transferring code rationales into natural texts for teaching MMLMs seems innovative, and makes sense.

3. The ablation study has emperically proved the effectiveness of three operations, which mostly resolves my confusions.

**Limitations:**

Overall I do not find great flaws of this paper. Some points to consider:
1. The paper emphasizes on the pipeline of generating high-quality rationales from Python code, while I think another important gap is how to integrate these rationales into MMLMs. The authors propose a doubly right objective by introducing an extra learning objective. I would like to know if there is other ways, such as taking as input. The authors do not discuss or compare these integration methods.

2. I do not quite get the motivation of symbolic merging. From my view it's just a kind of symbol conversion to present with an alternate form. It may be conducive to implementation but it seems does not prune redundant nodes.

**Suitability:**

2

---

### Official Review · Reviewer_KANi · 2024-05-21

**Rating:** 3
**Confidence:** 3

**Summary:**

This paper introduces a new paradigm called Fact, aiming at generating faithful, concise, and transferable multimodal rationales for multimodal large language models (MLLMs). The article explores the opaque reasoning process, limited combinatorial reasoning ability, and challenges posed by hallucination risk in MLLMs. Fact utilizes verifiable visual programming to generate faithful and precise reasoning, enhances simplicity through operations such as pruning and merging, and ensures transferability to end-to-end paradigms. Experimental evidence shows that Fact is effective in improving the compositional reasoning and generalization ability of MLLMs, while reducing hallucinations.

**Strengths:**

1. The paper is easy to understand and the structure is clear.
2. The experiment is rich and the results are excellent. After using Fact, MiniGPT4 and OpenFlamingo models have improved on multiple datasets.
3. The author proposes CoT Transferability Verification to explore whether the generated CoT is suitable for adapting to downstream models.

**Limitations:**

n the contribution, it is claimed that “We validate that the CoT rationale generated through a programming-based approach is applicable for distillation into end-to-end models.” In fact, Visual Program Distillation (VPD) has already been validated.
2. Why is the backbone value in Table 4 inconsistent with that in Table 1. After using COT data for fine-tuning, the performance actually decreased? Is it because there is no filtering based on whether the answer is correct? But if faithfulness is added, the method actually becomes VPD. Why is the performance still lower than the OpenFlamingo-Instruct (3B) generalist when selected CoTs are introduced? This does not match the experimental results of VPD, which is confusing.
3. Most complete programs actually possess logic, but the removal of specific nodes can result in subsequent logical gaps. From Figure 3, it can be seen that it is because you deleted nodes 3 and 4 that the subsequent logical gap occurred. Actually, nodes 3 and 4 are executed nodes, why should they also be deleted? Also, why do you need to fine tune an additional T5? Can't LLama2-70B supplement the logical gap by itself through prompts?
4. By the way, faithfulness is not considered innovative, as many methods have done so like VPD.
5. Did captions be used as input when generating programs with GPT-3.5. If so, how is the caption obtained, is it detailed or brief, and does it cause hallucinations?
6. In the Migrating to Large Models experiment, simply comparing whether CoT is used is not enough. It is also necessary to analyze whether the CoT generated by Fact is better for LLMs than the CoT generated by other methods in in-context learning .
7. When conducting Transferable Rational Verification, do we use the models that require further fine-tuning for verification, and then distill them separately (the distillation data used for the two models after verification is different). Or only perform one verification and use the same distillation data for OpenFlamingo and MiniGPT4.
8. There are some writing errors in the paper, such as missing a comma in line 514, and line 800 should be Table 3 instead of Table 2. What does this 'approach' demonstrate the efficiency of 'our method' mean in line 828?
9. Surprisingly, the template used is the 2017 version rather than the latest one.

**Suitability:**

2

---

### Official Review · Reviewer_c1QL · 2024-05-25

**Rating:** 3
**Confidence:** 3

**Summary:**

This article introduces Fact, a novel approach that focuses on producing faithful, concise, and transferable multimodal rationales for teaching MLLMs. Fact leverages verifiable visual programming to generate executable code. Additionally, through a series of operations such as pruning, merging, and bridging, the rationales are refined to enhance their conciseness. Moreover, Fact employs a filtering mechanism to identify rationales that can be seamlessly transferred from programming paradigms to end-to-end paradigms, thereby ensuring their transferability. Evaluation on tasks like GQA, OKVQA and COCO all reveal the effectiveness of the proposed method.

**Strengths:**

1. The paper introduces a novel paradigm to transform code-pretrained models’ logic into qualified rationale, improving the reasoning capabilities of MLLMs.
2. Three controllable editing operations (i.e., pruning, merging and bridging) are proposed to enhance the quality of COT rationales.
3. The ablation results provide sufficient evidence to validate the effectiveness of the proposed method.

**Limitations:**

1. More qualitative analysis is needed to better understand the advantages of the proposed method. Additionally, in Fig. 5, including results without Fact would facilitate a clearer understanding of the benefits brought by Fact.
2. The authors claim that the Fact paradigm effectively enhances the compositional and logical reasoning of MLLMs (Line 165-167). In that case, I think that more challenging and general benchmarks (e.g., MM-Bench and SEED) evaluations for MLLMs should be employed to assess their reasoning abilities, rather than just MME.
3. In Lines 512-513, the authors categorize the transferability of CoT rationales into three types (i.e., useful, non-useful, and unsure) by following GEN-U[1]. I'm wondering if there could be a more fine-grained classification (e.g., specifying the degree of usefulness) rather than a simplistic categorization of usefulness.

[1]  Are Machine Rationales (Not) Useful to Humans? Measuring and Improving Human Utility of Free-Text Rationales

**Suitability:**

2

---

### Meta-Review · Area_Chair_S9kr · 2024-07-03

**Recommendation:** Accept (Poster)
**Confidence:** 4

**Metareview:**

This paper presents an interesting method to enhance the reasoning ability of MLLMs by leveraging a code generation model to generate rationales and teach the MLLMs using the rationales. In the final ratings, two reviewers vote to accept, and one reviewer leans to reject. Most concerns are cleared in the rebuttal.  The reviewer who gave a 'borderline reject' mentioned that some of the concerns were addressed in the rebuttal but did not provide a strong reason for rejection in the final comment. The AC agrees with the majority and recommends acceptance.